# Transplantation of exogenous mitochondria mitigates myocardial dysfunction after cardiac arrest

Zhen Wang[1†], Jie Zhu[1†], Mengda Xu[2], Xuyuan Ma[3], Maozheng Shen[3], Jingyu Yan[2], Guosheng Gan[1,2]*, Xiang Zhou[1,2]*

[1]The First School of Clinical Medicine, Southern Medical University, Guangzhou, China; [2]Department of Anesthesiology, General Hospital of Central Theater Command of PLA, Wuhan, China; [3]Base of Central Theater Command of People's Liberation Army, Hubei University of Medicine, Wuhan, China

**\*For correspondence:**
526193186@qq.com (GG);
zhouxiang188483@126.com (XZ)

[†]These authors contributed equally to this work

**Competing interest:** The authors declare that no competing interests exist.

## eLife Assessment

In this **valuable** report, the authors investigated the effect of mitochondrial transplantation on post-cardiac arrest myocardial dysfunction (PAMD), which is associated with mitochondrial dysfunction. They **convincingly** demonstrated that mitochondrial transplantation enhanced cardiac function and increased survival rates after the return of spontaneous circulation (ROSC). They have also shown that myocardial tissues with transplanted mitochondria exhibited increased mitochondrial complex activity, higher ATP levels, reduced cardiomyocyte apoptosis, and lower myocardial oxidative stress post-ROSC.

**Abstract** The incidence of post-cardiac arrest myocardial dysfunction (PAMD) is high, and there is currently no effective treatment available. This study aims to investigate the protective effects of exogenous mitochondrial transplantation in Sprague-Dawley (SD) rats. Exogenous mitochondrial transplantation can enhance myocardial function and improve the survival rate. Mechanistic studies suggest that mitochondrial transplantation can limit impairment in mitochondrial morphology, augment the activity of mitochondrial complexes II and IV, and raise ATP level. As well, mitochondrial therapy ameliorated oxidative stress imbalance, reduced myocardial injury, and thus improved PAMD after cardiopulmonary resuscitation (CPR).

## Introduction

Cardiac arrest (CA) is a life-threatening event and that is followed by post-cardiac arrest myocardial dysfunction (PAMD) after the return of spontaneous circulation (ROSC). PAMD exhibits degrees of systolic and diastolic dysfunction. The heart requires up to 30 kg of ATP daily to maintain systolic function (*Hall et al., 2014*). Due to the high energy requirements of the myocardium, mitochondria make up approximately one-third of the volume of cardiomyocytes (*Cao et al., 2023*). The occurrence of PAMD is associated with mitochondrial damage with increased mitochondrial volume, accumulation of calcium ions, a diminution in complex activity, and a decrease in phosphate synthesis (*Huang et al., 2015*; *McCully et al., 2023*; *Su et al., 2021*; *Tsai et al., 2021*). Myocardium, as a high energy-consuming tissue, is very susceptible to energy depletion due to mitochondrial dysfunction. After mitochondrial injury, cytoplasmic and mitochondrial calcium overload prevents full relaxation of cardiomyocytes and can release cytochrome C to activate mitochondrial apoptotic pathways. This process contributes to the development of PAMD (*Gazmuri and Radhakrishnan, 2012*; *Su et al.,*

**Figure 1.** Experimental design of the in vivo study, rats in all groups except the Sham group underwent cardiac arrest for 5 min and then received the corresponding intervention 10 min after ROSC. Four hours after ROSC, the 4 hr group was used to collect myocardial tissue and blood samples for detection, while the 72 hr group was used for survival detection. BL, baseline; CA, cardiac arrest; ROSC, return of spontaneous circulation.

*2021*). Therefore, maintaining cardiomyocyte mitochondrial function after cardiopulmonary resuscitation (CPR) could be useful for cell survival and tissue homeostasis.

Transplantation of mitochondria isolated from non-ischemic or allogeneic tissues promoted myocardial recovery after regional ischemia (*McCully et al., 2009*). Although the safety and efficacy of this approach have been verified (*Ali Pour et al., 2021*; *Jia et al., 2022*), effectiveness of mitochondrial transplantation in treating myocardial injury following CA remains to be demonstrated. Herein, we performed mitochondrial transplantation in rats following CPR to observe its effects on cardiac function and prognosis and explored its potential mechanism of action *Figure 1*.

## Results

### CA-CPR model

No significant differences in weight or temperature were found between any groups before modeling. There was no significant difference in CA induction or CPR duration among the normal saline (NS), respiration buffer (Vehicle), and mitochondrial suspension (Mito) groups. Compared with the Sham group, the other three groups tended to receive less anesthetic (*Table 1*).

**Table 1.** Baseline characteristics of rats and resuscitation characteristics ($\bar{x}\pm s$).

| Variable | n | Weight (g) | Temp (°C) | CA-induced time (sec) | CPR time (s) | Adrenaline (mcg) | Pentobarbital sodium (mg) |
|---|---|---|---|---|---|---|---|
| Sham | 17 | 321.89±14.09 | 37.20±0.36 | - | - | - | 22.82±2.01 |
| NS | 17 | 318.65±10.00 | 37.45±0.43 | 234.29±22.89 | 128.41±12.50 | 12.75±0.40 | 21.63±1.35 |
| Vehicle | 17 | 326.80±11.64 | 37.33±0.37 | 247.06±23.61 | 138.82±14.64 | 13.07±0.47 | 21.79±0.98 |
| Mito | 17 | 320.64±14.32 | 37.45±0.39 | 241.29±23.38 | 131.88±14.51 | 12.83±0.57 | 21.92±1.05 |
| p-Value | | 0.29 | 0.21 | 0.57 | 0.10 | 0.13 | 0.07 |

n=17 animals per group. CA, cardiac arrest; CPR, cardiopulmonary resuscitation. All data are expressed as Mean ± SD.

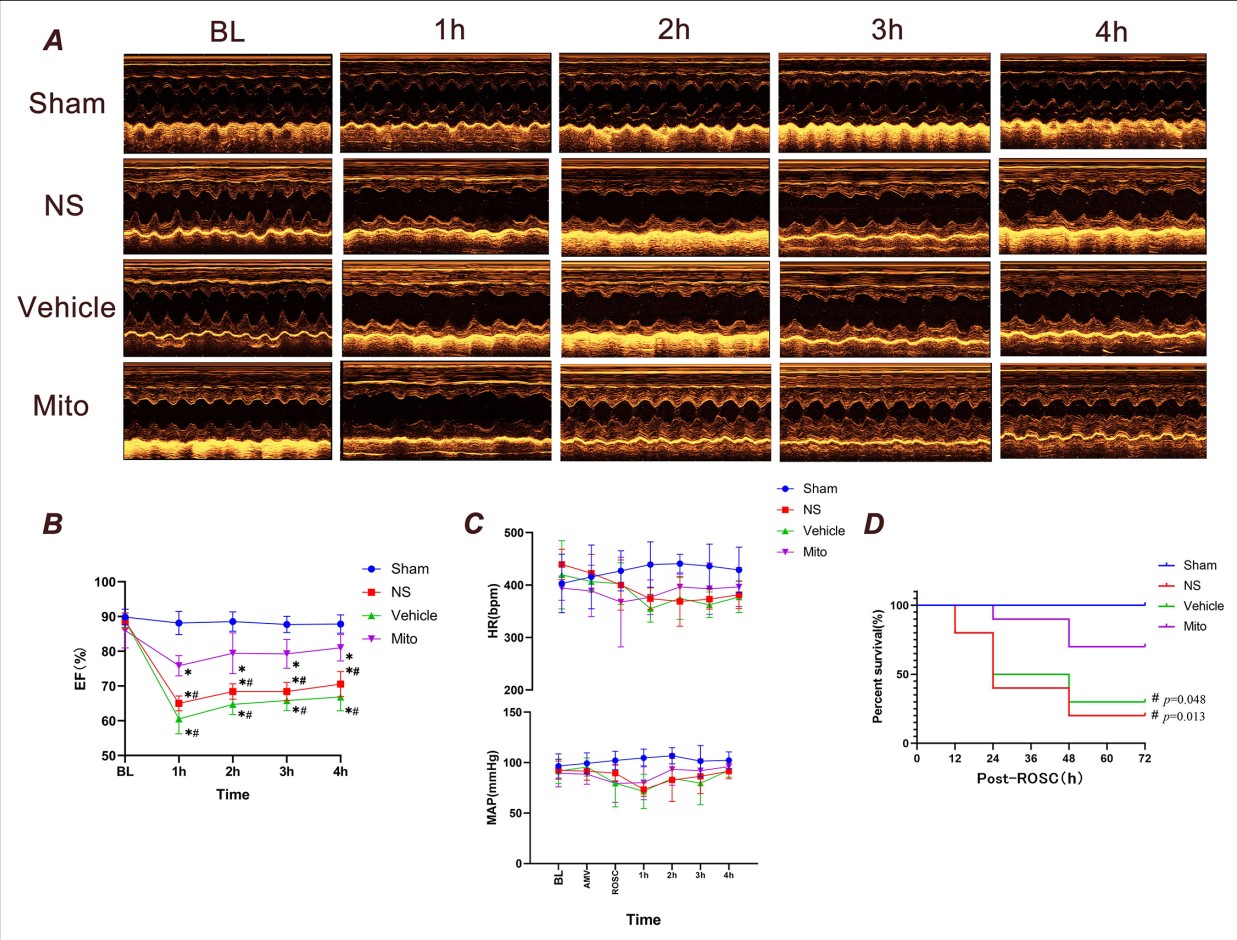

**Figure 2.** Observational results at 4 and 72 hr after cardiopulmonary resuscitation in rats. (**A**) Echocardiograms of rats in each group from baseline to 4 hr following ROSC (n=7). (**B**) EF of rats in each group from baseline to 4 hr following ROSC (n=7). (**C**) HR and MAP changes during post -ROSC in 4 hr (n=7). (**D**) Survival rate during the first 72 hr following ROSC (n=10). Data presented as mean ± standard deviation (SD). Myocardial function between groups was compared by time-based measurements in each group using repeated-measures ANOVA. The survival rate between groups was compared by the Kaplan-Meier survival analysis test. * p<0.05 vs. the Sham group and # p<0.05 vs. the Mito group. BL, baseline; EF, ejection fraction; HR, heart rate; MAP, mean arterial pressure; bpm, beats per minute; mmHg, millimeters of mercury; AMV, after mechanical ventilation; ROSC, return of spontaneous circulation.

## Mitochondrial transplantation improves cardiac function and does not alter hemodynamics in rats

Echocardiography revealed no differences in cardiac function among the groups prior to CA-CPR. Compared with the baseline, the left ventricular ejection fraction of rats subjected to CA-CPR significantly decreased at 1 hr after ROSC, and gradually improved over time (p<0.05; *Figure 2A and B*). Of note, the level of cardiac-function impairment in the Mito group was significantly reduced 4 hr after ROSC compared with the NS and Vehicle groups (p<0.05).

There was no statistically significant difference between MAP and HR at any observational time-points (p>0.05, *Figure 2C*). In the NS, Vehicle and Mito groups, the mean arterial pressure (MAP) and heart rate (HR) decreased at 1, 2, 3, and 4 hr after ROSC, reaching their nadir at 1 hr. Subsequently, MAP and HR increased gradually but did not show any statistically significant differences compared with the Sham group.

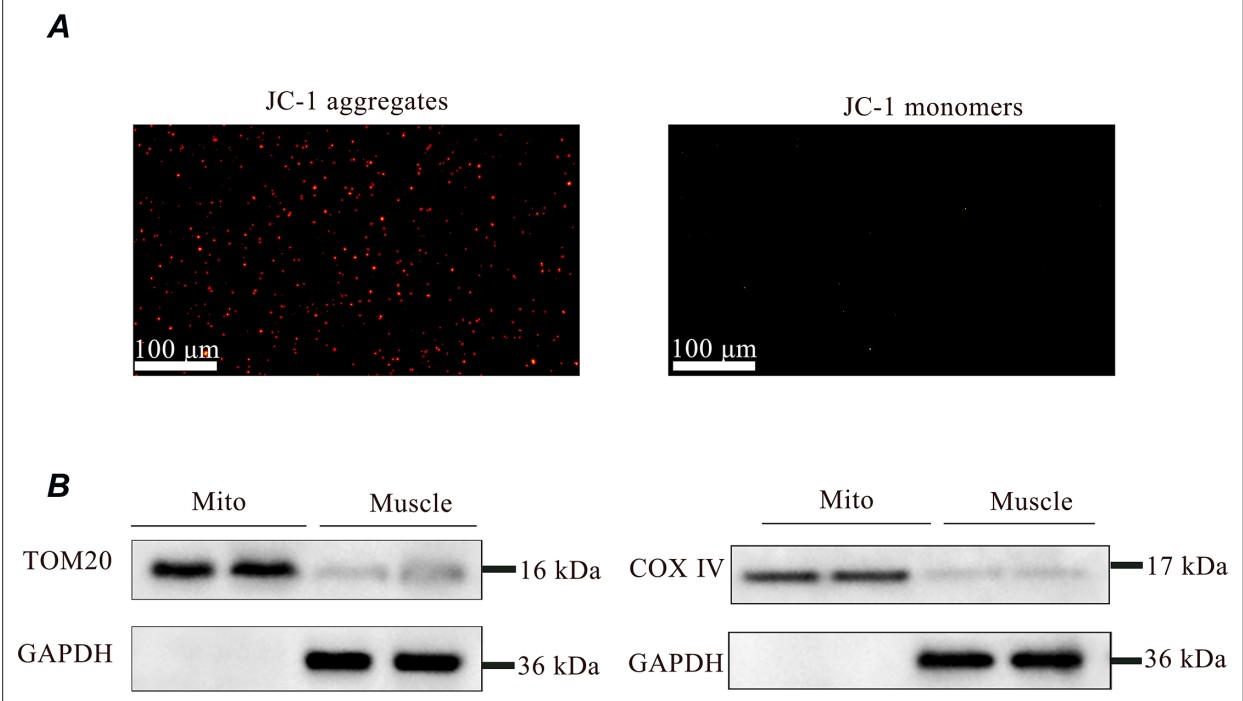

**Figure 3.** Assessment of the viability and purity of isolated mitochondria. (**A**) JC-1 staining of mitochondria after isolation from muscle. The staining of isolated mitochondria by JC-1 is visible either as red for J-aggregates or green for J-monomers. The intensity of the red color indicates that the isolated mitochondria had a high membrane potential, confirming their quality for transplantation. Scale bar = 100 μm. (**B**) SDS/PAGE analysis of fractions obtained during the purification of muscle mitochondria. GAPDH is only expressed in muscle, confirming its purity for transplantation.

The online version of this article includes the following source data for figure 3:

**Source data 1.** PDF file containing uncropped western blots with labeling for panel B.

**Source data 2.** Original tiff files of western blots for panel B.

## Mitochondrial transplantation improves the 72-hr survival rate after CPR in rats

In animals not treated with mitochondria, Kaplan-Meier analysis found a rapid drop in the survival rate of rats within 24 hr after ROSC. In contrast, the survival rate of the Mito group was significantly higher than that of the NS and Vehicle groups (p<0.05, *Figure 2D*).

## Viability and of isolated mitochondria

To evaluate the mitochondrial membrane potential (ΔΨm), JC-1 dye was used JC-1 dye aggregates in healthy mitochondria and fluoresces red. Upon the decrease of ΔΨm, JC-1 can only exists as monomers and fluoresces green. A more pronounced red fluorescence indicates a higher number of normal mitochondria. This finding suggests that the majority of mitochondria extracted from the gastrocnemius muscle can maintain normal mitochondrial function (*Figure 3A*). Mitochondria purity was confirmed by immunoblotting for TOM20 (a mitochondrial-specific protein) and COX IV in both skeletal samples and isolated mitochondrial proteins. GAPDH was only detected in skeletal muscle proteins (*Figure 3B*).

## Effects of mitochondrial transplantation on cardiomyocyte mitochondria

### Transplanted Mitochondria are internalized by cardiomyocytes

To verify the transfer of mitochondria to cardiomyocytes, we labeled the mitochondria with Mito-Tracker Red and transplanted them into the myocardium via the femoral vein. We primarily seek evidence of mitochondrial internalization within the endocardium, as initial injury occurs in this region during myocardial ischemia (*Kuwada and Takenaka, 2000*). Four hours after injection, analysis of

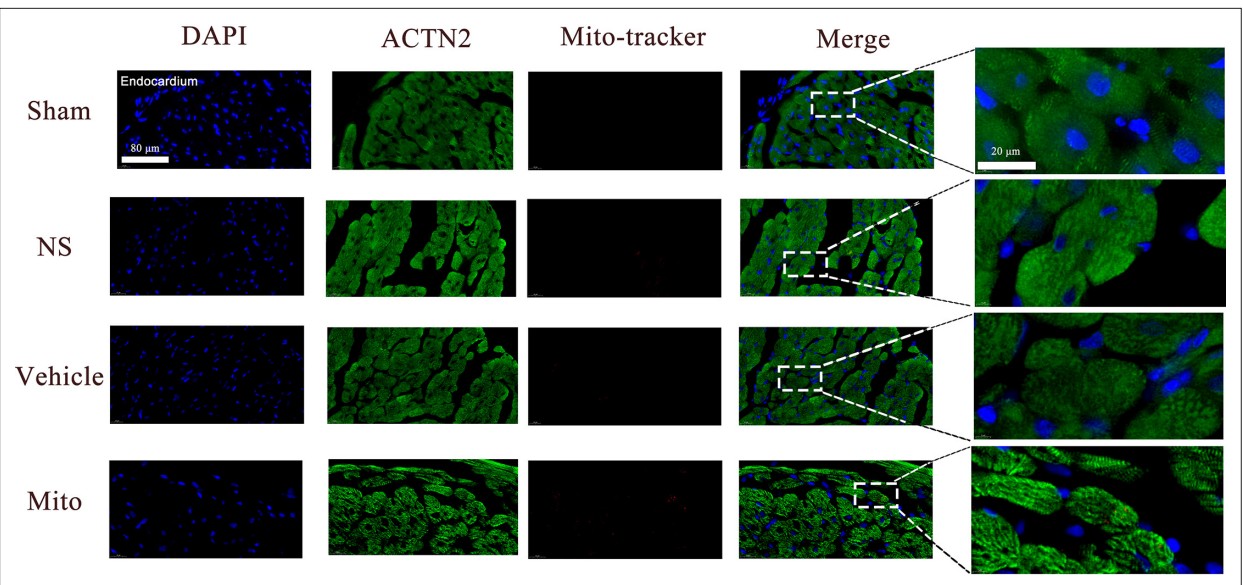

**Figure 4.** Localization and uptake of transplanted mitochondria in endocardium, myocardial tissue was stained for anti-α-actinin 2 (ACTN2; green) and nuclei (blue); the pre-stained isolated mitochondria were labeled red (n=3). Scale bar = 80 μm.

The online version of this article includes the following figure supplement(s) for figure 4:

**Figure supplement 1.** Localization and uptake of transplanted mitochondria in myocardium and epicardium, myocardial tissue was stained for anti-α-actinin 2 (ACTN2; green) and nuclei (blue); the pre-stained isolated mitochondria were labeled red (n=3).

tissue sections found labeled mitochondria in endocardium (*Figure 4*). The uptake of exogenous mitochondria was also observed in the myocardium and epicardium (*Figure 4—figure supplement 1*).

## Mitochondrial transplantation does not alter the mass of myocardial mitochondria

The TOM20 protein is constitutively expressed in the outer membrane of mitochondria, and its abundance correlates with the amount of mitochondrial mass (*Abrigo et al., 2023*). To investigate the potential increase of mitochondrial mass in myocardial tissue through mitochondrial transplantation, the expression of TOM20 was assessed in each group using tissue immunofluorescence staining. After 4 hr of ROSC, there was no significant difference in TOM20 level (as reflected by fluorescence intensity) among all groups, suggesting that mitochondrial transplantation did not increase the mass of mitochondria in heart. (*Figure 5A and B*).

## Mitochondrial transplantation improves cardiac mitochondrial structure

Transmission electron microscopy (TEM) micrographs of heart tissues were presented in *Figure 5C*. By using the Flameng method, the extent of mitochondrial damage in heart tissues across the four groups was assessed and the result was presented in *Figure 5D*. A higher score indicates more severe injury. The scores in the NS and Vehicle groups were significantly higher than those in the Mito group (p<0.05), indicating that mitochondrial damage in the Mito group was mitigated by mitochondrial transplantation. In the NS and Vehicle groups, the myocardial myofibrils exhibited a disordered arrangement on TEM, with variation in sarcomeres length and loss of Z-lines. Translucent and swollen mitochondria were visible, exhibiting a larger gap between membranes, distorted cristae, and accumulated calcium. In contrast, the mitochondria sarcomeres of the Mito group were more distinct, and the myocardial myofibrils were arranged in a more orderly fashion. The M-lines and Z-lines were clearly visible, and the number of mitochondrial cristae had increased, with their structure appearing to be complete.

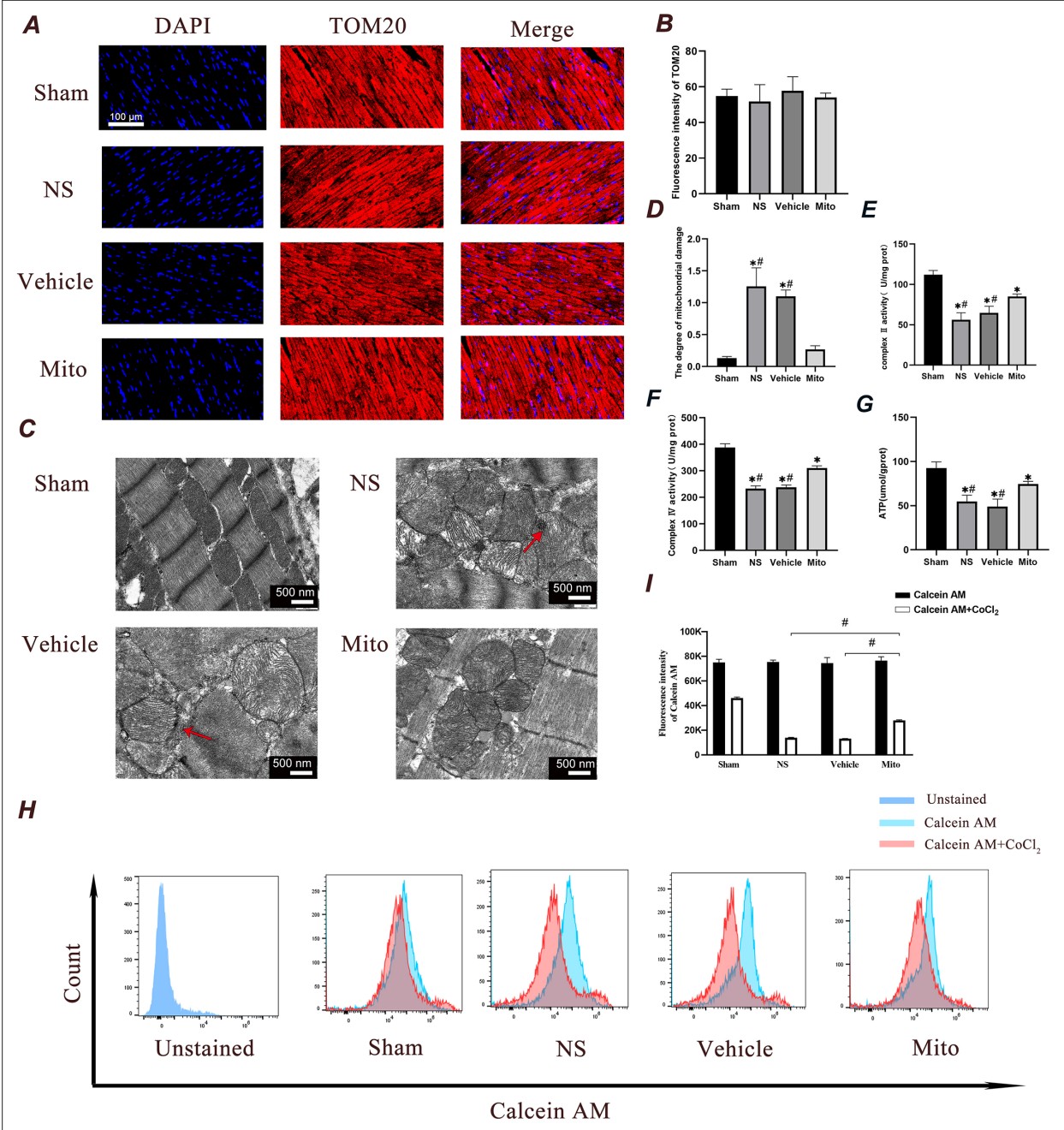

**Figure 5.** Administration of mitochondria ameliorates ischemia reperfusion-mediated mitochondrial alterations in cardiomyocytes four hours after ROSC. (**A and B**) A and B represent the detection of fluorescence intensity of TOM20. A represents the image, while B represents the quantitative data. Scale bar = 100 μm (n=3). (**C and D**) C shows representative photographs of mitochondrial morphology obtained through TEM examination, with arrows indicate calcium accumulation, D illustrates the degree of mitochondrial damage (n=3). Scale bars = 500 nm. (**E and F**) Changes in myocardial mitochondrial complex II and IV enzyme activities in hearts (n=7). (**G**) The ATP content in myocardial tissue was measured by colorimetry (n=7). (**H and I**) mPTP opening was detected by Calcein staining. I represent the quantitative analysis of the mean fluorescent intensity acquired in H (n=3). Analyses were performed using ANOVA with Tukey's post hoc test. mPTP opening detection between groups was compared by time-based measurements in each group using repeated-measures ANOVA. The data were expressed as the mean ± standard deviation (SD). * p<0.05 vs. the Sham group and # p<0.05 vs. the Mito group.

## Mitochondrial transplantation improves mitochondrial metabolism of cardiac tissue after CA-CPR

The mitochondrion supplies nearly all of the cell's energy through oxidative phosphorylation, which occurs via the mitochondrial respiratory chain. In instances of global ischemia, there is a deficiency of oxygen available to the mitochondria, which inhibits oxidative phosphorylation (*Rousou et al., 2004*). The activity of mitochondrial complexes II and IV, along with ATP content in cardiac tissue, were assessed in our research to reflect mitochondrial metabolism. Compared with the Sham group, complex II and IV activity and ATP content were significantly attenuated in the other three groups ($p < 0.05$). In contrast, the activity of complexes II and IV and ATP content were significantly elevated in the Mito group compared with the NS and Vehicle groups, suggesting that transfer of exogenous mitochondria enhanced function and energy metabolism of resident mitochondria ($p < 0.05$, *Figure 5E, F and G*).

## Mitochondrial transplantation limits mitochondrial permeability transition pore (mPTP) opening

The opening of the mPTP induces mitochondrial permeability transition, resulting in the uncoupling of oxidative phosphorylation, an increase in mitochondrial volume (swelling), and the dissipation of the mitochondrial membrane potential. Following calcein acetoxymethyl ester (Calcein AM) staining and cobalt chloride ($CoCl_2$) treatment, the opening level of mPTP was assessed using flow cytometry. In this assay, stronger fluorescence intensity indicates lower mPTP opening (*Figure 5H and I*). The rats in the Mito group demonstrated lower fluorescence intensity, which means a higher mPTP opening level compared with the Sham group. The rats of NS and Vehicle groups showed further increased mPTP opening level compared with the Mito group. Mitochondrial transplantation caused a significant reduction in mitochondrial swelling.

## Effects of mitochondrial transplantation on cardiomyocytes

### Mitochondrial transplantation alleviates myocardial oxidative stress injury after CA-CPR

Increased level of reactive oxygen species (ROS) during reperfusion are a critical factor in the development of reperfusion injury. Consequently, oxidative stress in the ischemic myocardium of rats undergoing CA-CPR was evaluated. First, we evaluated the level of ROS in myocardial mitochondria. We used probe DCFH-DA, which releases green fluorescence in response to esterase oxidation, to measure ROS level based on fluorescence intensity. As shown in *Figure 6A*, the ROS content in the myocardial mitochondria increased after 4 hr of reperfusion, and decreased with the transplantation of isolated mitochondria. Second, the production of malondialdehyde (MDA), which is widely used as 'footprints' of ROS generation, was also monitored. MDA level in the myocardial tissue of rats 4 hr after ROSC was decreased in animals given exogenous mitochondria (*Figure 6B*, $p < 0.05$). Third, our results indicated that the activity of superoxide dismutase (SOD) was significantly elevated in the Mito group compared with both NS and Vehicle groups (*Figure 6C*, $p < 0.05$).

### Mitochondrial transplantation reduces cardiomyocyte apoptosis

Apoptotic indices and the apoptotic rate, as assessed by the terminal deoxynucleotidyl transferase dUTP nick-end labeling (TUNEL) assay and flow cytometry, increased 4 hr after ROSC. Compared with the NS and Vehicle groups, the apoptotic indices and apoptotic rate in the Mito group were lower ($p < 0.05$), which linked exogenous mitochondrial transplantation and cardiomyocyte apoptosis reduction after CPR (*Figure 6D, E and F*). The expression of cleaved caspase-3 protein in the Mito group was lower than that in the NS and Vehicle groups. (*Figure 6G and H*).

### Mitochondrial transplantation reduces the markers of myocardial injury

ELISA was used to detect the expression of myocardial injury markers, Creatine Kinase MB isoenzyme (CK-MB) and Cardiac Troponin I (cTn-I), at 4 hr after ROSC. Compared with the Sham group, the level of CK-MB and cTn-I in the serum of the other three groups showed a significant rise ($p < 0.05$). However, both CK-MB and cTn-I level in the serum of the Mito group were lower than those in the NS and Vehicle groups, suggesting that myocardial injury was reduced ($p < 0.05$, *Figure 6I and J*).

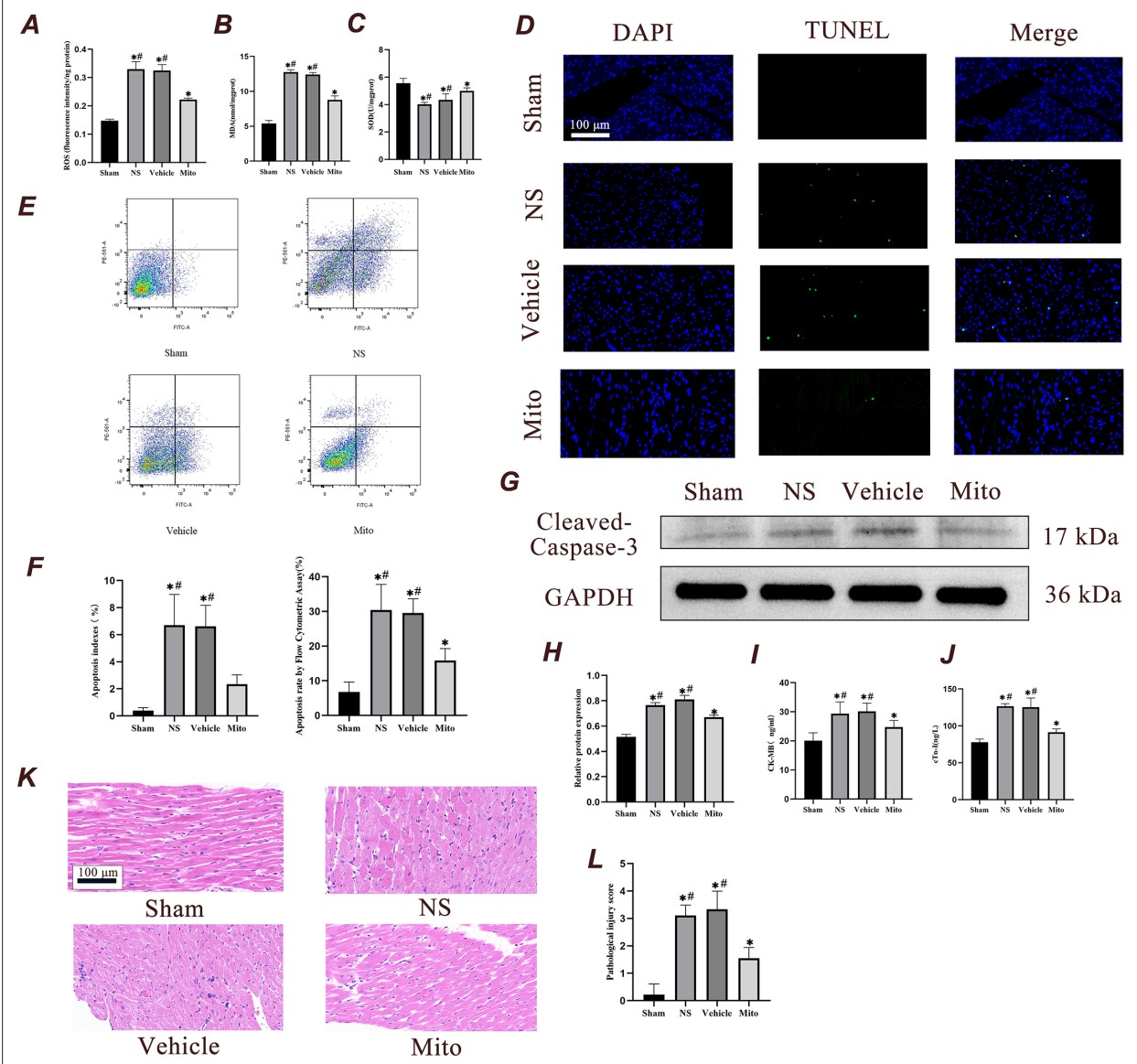

**Figure 6.** Mitochondrial transplantation reduces myocardial damage 4 hr after ROSC. (**A**) Detection of mitochondrial ROS in various groups (n=3). (**B and C**) Changes in malondialdehyde and superoxide dismutase activity level in cardiac tissue (n=7). (**D**) Myocardial apoptosis level was examined using TUNEL (n=3). Scale bar = 100 μm. (**E**) The percentage of myocardial apoptosis was examined using flow cytometry (n=3). (**F**) Quantitative analysis of myocardial TUNEL apoptosis index and flow apoptosis rate (n=3). (**G and H**) Immunoblotting and quantitative analysis of the expression level of cleaved caspase-3 in the myocardium 4 hr after ROSC (n=3). (**I and J**) The changes in CK-MB and cTn-I level in the serum of rats were examined using ELISA (n=7). (**K and L**) Representative histological sections of the myocardium stained with hematoxylin and eosin. Myocardium from each experimental group were subjected to histological evaluation (n=3), Scale bar = 100 μm. Analyses were performed using ANOVA with Tukey's post hoc test. The data were expressed as mean ± standard deviation (SD). * $p<0.05$ vs. Sham group and # $p<0.05$ vs. Mito group. CK-MB, creatine kinase-MB fraction; cTn-I: cardiac troponin-I.

The online version of this article includes the following source data for figure 6:

**Source data 1.** PDF file containing uncropped western blots with labeling for panel G.

**Source data 2.** Original tiff files of western blots for panel G.

## Mitochondrial transplantation attenuates myocardial histopathologic changes

Four hours after ROSC, myocardium from each experimental groups were subjected to histological evaluation (*Figure 6K and L*). Tissue samples from the Sham group exhibited a well-organized

structure of myocardial fibers, with no apparent inflammatory cell infiltration into the stroma. In the NS and Vehicle groups, the myocardial fibers were disordered, with some myocardial cells exhibiting thickening, rupture, and vacuolization. Also, there was inflammatory cell invasion in the stroma. The myocardial arrangement in the Mito group was more neatly arranged than that in the NS and Vehicle groups, although thickening of myocardial cells was noted.

## Discussion

PAMD is a common complication following resuscitated. It is characterized by systolic and diastolic insufficiency, arrhythmias, and recurrent CA. Circulatory dysfunction occurs in approximately two-thirds of CA subjects, which is associated with a poor outcome (*Han et al., 2022*). Unlike myocardial depression caused by regional ischemia characterized by localized ventricular wall-motion abnormalities, myocardial depression caused by PAMD after CA manifest as global left ventricular systolic dysfunction accompanied by reduced ejection fraction, left ventricular diastolic dysfunction, and right ventricular dysfunction (*Lazzarin et al., 2022*; *Xu et al., 2008*). Of these, global left ventricular systolic dysfunction is the most significant manifestation.

In the present study, we tested the hypothesis that the administration of exogenous mitochondria is cardioprotective effects after CPR (*Figure 7*). The rationale was to enhance the energy-dependent contractility of the myocardium. We isolated mitochondria from allogeneic gastrocnemius muscle tissue of healthy rats and maintained optimal mitochondrial activity and therapeutic effects. CA is an occasional event, making it impossible to predict when it will occur. Consequently, it is consistent with clinical practice to perform mitochondrial transplantation interventions only after spontaneous circulation has recovered and stabilized. The dosage of mitochondria is determined based on previous studies (*Blitzer et al., 2020*; *Guariento et al., 2020*). In previous research, isolated mitochondria $(1 \times 10^9)$ were delivered to the left coronary ostium in pigs, and can be a viable treatment in cardiac ischemia-reperfusion injury (*Blitzer et al., 2020*; *Guariento et al., 2020*). Additionally, the dose of $1 \times 10^9$ mitochondria achieves the maximal hyperemic effect when administered via intracoronary injection (*Shin et al., 2019*). Considering that Sprague-Dawley (SD) rats are smaller than pigs and that there is a loss of mitochondria during pulmonary circulation, we adopted a mitochondrial transplantation dose of $5 \times 10^8$. Subsequently, uptake of the delivered mitochondria occurred in the heart and myocytes, perhaps via micropinocytosis (*Kami and Gojo, 2020*). AS well, mitochondrial internalization occurred through actin-dependent endocytosis (*Pacak et al., 2015*). Internalized mitochondria promote degradation of resident via lysosomes to maintain an energy mitochondrial balance (*Jia et al., 2022*; *Liu et al., 2022*). Notably, (*Cowan et al., 2017*) indicated that isolated mitochondria are transported to endosomes and lysosomes; however, most of these mitochondria escape from these compartments and subsequently fuse with the endogenous mitochondrial network. Internalized mitochondria facilitated respiratory restoration in recipient cells with depleted mtDNA, and improved cellular function by increasing ATP level and oxygen-consumption rates (*Pacak et al., 2015*).

In our results, increased numbers of mitochondria were not found in the hearts of treated animals. Possibly because CPR caused ischemia/reperfusion injury to organs throughout the body, with non-specific distribution and uptake of the delivered mitochondria. Moreover, mitochondria possess the ability to adjust their mass and function to maintain cellular homeostasis. This includes mitochondrial dynamics such as fusion and fission, as well as biogenesis and degradation through mitophagy, which are crucial for regulating the morphology, mass, and function of mitochondria (*Quiles and Gustafsson, 2020*). These mechanisms may regulate the continuous remodeling of mitochondrial numbers to maintain mitochondrial mass homeostasis following mitochondrial transplantation (*Jia et al., 2022*). After mitochondrial transplantation, the energy supply of myocardial mitochondria was restored, and autophagy was enhanced, resulting in an increased clearance of damaged mitochondria (*Xu et al., 2024*). This also clarifies why the overall mass of mitochondria did not increase. *Masuzawa et al., 2013* proposed that the internalization of mitochondria is not critical, as cardioprotection occurs rapidly within 10 min of reperfusion, the presence of viable mitochondria in cardiac tissue is all that is required. The nonviable mitochondria, mitochondrial fractions, mitochondrial deoxyribonucleic acid, ribonucleic acid, and exogenous adenosine diphosphate and ATP are incapable of providing protection to the ischemic heart (*Masuzawa et al., 2013*). *Shin et al., 2019* also suggested that the number of mitochondria required for cardioprotection is not dependent on the absolute number of transplanted mitochondria. Therefore, an increase in mitochondrial mass may not be the mechanism

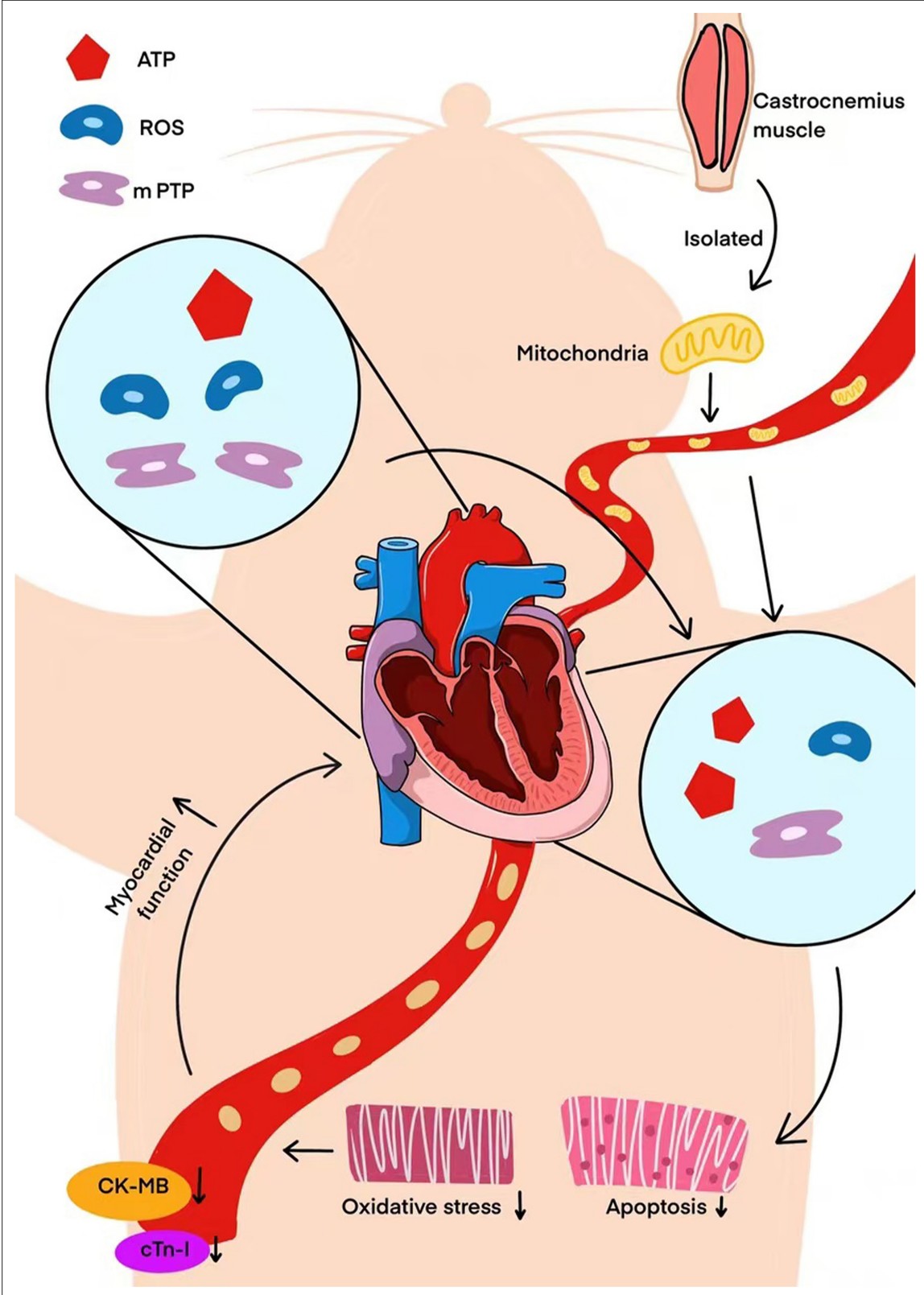

**Figure 7.** Exogenous mitochondrial transplantation improved cardiac function after CPR. The specific mechanism involved may be related to the improvement in mitochondrial function, thus reducing the oxidative-stress response and apoptosis of myocardial cells. These dates suggest possible advantage in mitochondrial transplantation following CPR.

underlying the cardioprotective effects of transplanted mitochondria. The cardioprotective mechanisms of isolated mitochondria involve increased ATP production. The results of our study were consistent with other date that the energy metabolism of myocardial mitochondria improved after mitochondrial transplantation along with increased complex activity and ATP content (*Jia et al., 2022*; *Pacak et al., 2015*). Mitochondrial transplantation is being pursued as a therapeutic strategy, as it does not result in coronary occlusion, autoimmunity and arrhythmia attacks (*Masuzawa et al., 2013*; *Shin et al., 2019*). The clinical application of mitochondrial transplantation can enhance the prognosis of pediatric patients requiring extracorporeal membrane oxygenation following cardiac surgery, the improvement of cardiac function in patients helps them successfully wean off ECMO support (*Emani and McCully, 2018*). It is worth noting that the method of mitochondrial isolation may better meet the needs of clinical settings where an interventional time below 60 min is desired (*Ali Pour et al., 2021*). After the isolation, mitochondria can be immediately used for transfer and internalization (*Ali Pour et al., 2021*). Still concerns regarding mitochondrial hemocompatibility and stability in serum have been raised. In this line, serum calcium did not adversely impact isolated mitochondria (*Maleki et al., 2023*).

ROS is generated with ischemia and then targets the normal physiologic functions of cells by damaging the structures of proteins, lipids, and even DNA (*Zhu et al., 2024*). As well, oxidative stress promotes cardiac ischemia-reperfusion injury and was implicated in pro-inflammatory responses and apoptosis, both of which worsen myocardial cell injury (*Patel and Karch, 2020*). In a regional ischemia-reperfusion model, mitochondrial transplantation reduced oxidative stress in cells following reperfusion (*Zhang et al., 2019*). Similarly, in the present study, mitochondrial transplantation reduced the production of ROS, decreased the level of myocardial oxidative markers, and increased the activity of antioxidant enzymes after 4 hr of ROSC.

Ischemia/reperfusion injury of myocardial tissue releases CK-MB and cTn-I, which were commonly used indicators of early myocardial injury in clinical settings (*Zhang et al., 2021*). The concentrations of CK-MB and cTn-I rose significantly 4 hr after resuscitation in the CA-CPR model, while mitochondrial transplantation significantly reduced this effect. During ischemia, the loss of mitochondrial membrane potential leads to mitochondrial swelling and release of cytochrome C, followed by caspase-mediated cell death during reperfusion (*Lu et al., 2023*; *Milliken et al., 2022*). Apoptosis is the process by which cardiomyocytes die (*Kunapuli et al., 2006*). Increase apoptosis was found in rats after CA-CPR along with increased caspase-3 expression after resuscitation (*Wu et al., 2021*). Similar results were reported in pigs that underwent CA-CPR (*Wang et al., 2023*). In our results, the level of apoptosis and the opening of mPTP in cardiomyocytes increased significantly after CPR, whereas administration of healthy mitochondria significantly reduced both apoptosis and mPTP opening. Moreover, exogenous mitochondrial transplantation significantly reduced the disordering of myocardial fiber arrangement and inflammatory cell infiltration.

In conclusion, exogenous mitochondrial transplantation improved cardiac function after CPR. The specific mechanism involved may be related to the improvement in mitochondrial function, thus reducing the oxidative-stress response and apoptosis of myocardial cells. These dates suggest possible advantage in mitochondrial transplantation following CPR.

## Limitations

Hemodynamic observation was carried out only for 4 hr after ROSC, Thus, the impact of mitochondrial transplantation on long-term cardiac function following CPR could not be assessed. In this study, the animal model consisted of healthy rats. However, in clinical practice, CA is primarily caused by cardiac issues. Healthy rats tend to exhibit a higher recovery ability after a CA attack, which may lead to an overestimation of the therapeutic effects of mitochondrial transplantation. In our next phase, we plan to utilize various CA models to conduct comprehensive studies to validate the effectiveness of mitochondrial transplantation in different pathological models.

## Methods
### Animals

The experimental protocol (*Figure 1*) was approved by the Animal Experiment Committee of the General Hospital of Central Theater Command (No.2023017) and conformed to the Guide for the

Care and Use of Experimental Animals published by the National Institutes of Health, USA (NIH Publication No. 5377–3, 1996). Adult male SD (7–8 weeks old) rats weighing 250–350 g were purchased from Hunan Silaikejingda Experimental Animal Co., Ltd. (No. 430727231102711675, Changsha, China). Male rats were selected to avoid estrous cycle interference. Animals were housed in standard cages with a 12 hr light-dark cycle at a room temperature 22 °C.

Animals were randomly assigned to four groups using the random number table method: a Sham group, NS group, Vehicle group, and Mito group (n=17). Each group was further divided into two survival-analysis subgroups: a 4-hr group (n=7) and a 72-hr group (n=10). Rats in the NS, Vehicle and Mito groups underwent CA induced by asphyxia and CPR, while rats in the Sham group underwent tracheal intubation and arteriovenous puncture only, without CA and CPR.

## Experimental protocol
### CA-CPR model
Establishment of the CA-CPR model was as previously published (*Huang et al., 2020*). Rats were anesthetized with pentobarbital sodium (45 mg/kg) via intraperitoneal injection. A 16-G intravenous catheter was used for endotracheal intubation. The left femoral artery and right femoral vein were subsequently inserted using a 24-G venous indwelling needle for dynamic blood pressure detection and venous access establishment. Blood pressure, HR, and rectal temperature were monitored using the ALC-MPA monitoring system (Alcott Biotech, Shanghai, China). After 10 min of mechanical ventilation, asphyxia-mediated CA was induced by blocking the tracheal tube. CA was defined as a decrease in systolic blood pressure to 25 mmHg. Five minutes after CA, external chest compression (200 bpm) and mechanical ventilation (tidal volume, 0.60 mL/100 g, respiratory rate, 80 bpm) were initiated, and epinephrine (40 mcg/kg) was administered intravenously. ROSC was defined as the spontaneous restoration of sinus rhythm with a systolic blood pressure of >60 mmHg that was maintained for 10 min. Subsequently, corresponding treatments were administered. Animals that did not regain autonomic circulation within 5 min were excluded from the study. After ROSC, hemodynamics and echocardiograms were recorded for 4 hr. Basal body temperature was maintained at 37 °C throughout the intervention. Pentobarbital sodium (10 mg/kg) was given if animals urinated or showed increased limb-muscle tension. Twenty-eight animals of 4 hr group completed the full protocol and were humanely euthanized with excess pentobarbital sodium 4 hr after ROSC. Myocardial tissue and blood samples were collected for detection. The remaining 40 animals of 72 hr group were monitored for 72 hr to track survival.

### Intervention measures
Ten minutes after ROSC, 0.5 mL of normal saline, 0.5 mL of respiration buffer, and 0.5 mL of $1 \times 10^9$/mL mitochondrial suspension were injected through the femoral vein in the NS, Vehicle, and Mito groups, The Sham group received 0.5 mL of NS at 30 minutes postoperatively.

## Echocardiography
To assess myocardial function, an ultra-high resolution, small-animal ultrasound imaging system (SigmaVET, Esaote, Genoa, Italy) was used to capture M-mode echocardiographic images at the parasternal long axis and/or parasternal short axis near the papillary muscle at 1 hr, 2 hr, 3 hr, and 4 hr post-ROSC.

## Survival analysis
In the 72 hr survival study cohort, the catheters were extracted 4 hr post-ROSC, and the surgical incisions were closed. Animals were placed back in their cages and administered subcutaneous injections of 0.5 mL of 1% lidocaine for analgesia. Animals were monitored every 2 hr during the initial 24 hr following ROSC, and then at 48 and 72 hr. Animals that exhibited signs such as wheezing or a respiratory rate below 5 breaths per minute were deemed moribund and humanely euthanized (*Su et al., 2021*). At the end of the time rats were euthanized.

## Isolation and identification of mitochondria

### Mitochondrial isolation and labeling

Mitochondria were isolated as published (*Preble et al., 2014*). In brief, 180 mg of gastrocnemius muscle from healthy male SD rats aged 7–8 weeks were placed into a centrifuge tube containing 5 mL of pre-chilled homogenizing buffer (300 mM sucrose, 10 mM K-HEPES, and 1 mM K-EGTA), homogenized, and incubated on ice with *Bacillus subtilis*, a protease (P5380, Sigma-Aldrich, Darmstadt, Germany), for 10 min. The mixture was filtered through a 40- μm cell filter, and then bovine serum albumin (ST023-50 g, Beyotime, Shanghai, China) was added and incubated for 5 min. The homogenate was then filtered through a 40-μm screen and then through a 10-μm screen, and centrifuged at 9000 × *g* for 10 min at 4 °C. The precipitate was reconstituted in 1 mL of pre-chilled respiration buffer (250 mM sucrose, 2 mM potassium dihydrogen phosphate, 10 mM magnesium chloride, 20 mM K-HEPES, and 0.5 mM K-EGTA). Mitochondrial amounts were determined as previously published and were quantified using a bacterial counter (0650010, Marienfeld, Lauda-Königshofen, Germany; *McCully et al., 2009*). The isolated mitochondria were incubated with Mito-Tracker Red CMXRos (C1049B-50μg, Beyotime, Shanghai, China) at 37 °C for 15 min and then washed three times with the mitochondrial respiration buffer.

### Detection of mitochondrial membrane potential

A mitochondrial membrane potential detection kit (JC-1; C2003S; Beyotime, Shanghai, China) was used to characterize function. Purified mitochondria were incubated with JC-1 staining solution at 37 °C for 20 min, followed by imaging with a fluorescence microscope (BX53, OLYMPUS, Tokyo, Japan) equipped with a WHN 10/22 eyepiece. The image analysis software used was OLYMPUS cellSens Standard version 3.2.

## Western blotting

Tissue samples and isolated mitochondria were lysed in RIPA buffer (P0013B; Beyotime, Shanghai, China) containing 1% PMSF (ST505; Beyotime, Shanghai, China) on ice to extract proteins. Protein concentration was determined by the BCA method. Protein was boiled in a water bath for 10 min. Electrophoresis was performed using a 12% SDS-PAGE gel (P0012A, Beyotime, Shanghai, China), followed by transfer to a PVDF membrane (FFP70, Beyotime, Shanghai, China). Non-specific binding was blocked with 5% skim milk powder at room temperature for 2 hr. Membranes were exposed to primary antibodies overnight at 4 °C: anti-Caspase-3 (1:1000, 14,220T, Cell Signaling Technology, Beverly, MA, USA), anti-TOM20 (1:1000, AF5206; Affinity, Changzhou, China), anti-COX IV (1:1000; AC610, Beyotime, Shanghai, China) and anti-GAPDH (1:3000; LF206, Epizyme, Shanghai, China) on a shaker at 4 °C for 12 hr. After washing, membranes were incubated with horseradish peroxidaselabeled secondary antibody (1:2000, A0208, Beyotime, Shanghai, China) for 2 hr and developed with the ECL chemiluminescence method. Results from immunoblots were scanned using the chemiluminescence imaging system (SH-523, Shenhua, Hangzhou, China; JY-Clear ECL, JUNYI, Beijing, China). The relative band intensities were quantified using ImageJ software (version 1.54 f; National Institutes of Health, USA; *Supplementary file 1*).

## Mitochondrial distribution and uptake

Mitochondrial co-localization was determined as published (*Masuzawa et al., 2013*). Tissue sections were incubated with 0.25% Triton-X-100 for 10 min at room temperature, blocked with normal bovine serum albumin, and incubated with primary antibody generated against alpha-actinin 2 (1:500; GTX103219; GeneTex, Irvine, USA) at 4 °C overnight. After washing with PBST, sections were incubated with an Alexa Fluor 488-labeled secondary antibody (1:1000, HZ0176; Huzhen, Shanghai, China) for 1 hr. DAPI (P0131-5 ml; Beyotime, Shanghai, China) was applied to stain nuclei. Images were acquired using the OLYMPUS fluorescence microscope.

## Analysis of cardiac mitochondria

### Complex activity

Mitochondria were extracted from 0.1 g myocardial tissue using differential centrifugation. The mitochondrial suspension was disrupted by ultrasonic waves in an ice bath releasing the

mitochondrial respiratory chain complex. The working solution from mitochondrial complex activity test kit (AKOP008M, AKOP006M; Boxbio, Beijing, China) was mixed with the samples and the absorbance at 605 nm and 550 nm measured values at various timepoints. The variance between the two timepoints was calculated and recorded as $\Delta A1$ and $\Delta A2$. Protein concentration (C) of each sample was measured. Mitochondrial complex II activity was calculated as $476.19 \times \Delta A1/C$, and mitochondrial complex IV activity was calculated as $1099.48 \times \Delta A2/C$.

## ATP determination
ATP level was determined using an ATP-content test kit (A095-1-1; Nanjing Jiancheng, Nanjing, China).

## TOM20 labeling
Tissue sections were labeled with TOM20 antibody (1:100, AF5206; Affinity, Changzhou, China), incubated with a horseradish peroxidase-coupled secondary antibody (1:400, 5220–0336; SeraCare, Massachusetts, USA), subjected to tyramide signal amplification (TSA), stained with DAPI and images acquired using an OLYMPUS fluorescence microscope.

## Transmission electron microscope
Fixed hearts were rinsed with phosphate-buffer ed saline (PBS) for 45 min and exposed to 1% osmium tetroxide for 2 hr. After dehydration in graded ethanol concentrations, tissue samples were immersed in epoxy resin, embedded, and cut into ultrathin sections (60–80 nm). The sections were stained with uranium and lead, and examined and imaged using a TEM (HT7800; Hitachi, Tokyo, Japan). The degree of mitochondrial damage in the four groups was assessed using the Flameng classification method. The evaluation is conducted by trained experimenters who are unaware of the group assignments. Mitochondrial damage was evaluated on a scale from 0 to 4, where 0 indicates a normal structure, 1 indicates normal with slight swelling, 2 indicates mitochondrial swelling, 3 indicates serious swelling and cristae disorder, 4 indicates mitochondria membrane breach and vacuolization. Based on this scale, the degree of mitochondrial damage in rats across the four groups was scored, and the average scores were calculated from three random fields of view per rat. A higher score indicates more severe injury (*Zhang et al., 2017*).

## The determination of ROS
The extraction of mitochondria from myocardial tissue was performed using a mitochondrial isolation kit (S3606; Beyotime, Shanghai, China). Rat myocardial tissues were homogenized with mitochondrial isolation reagent A and then centrifuged at $600 \times g$ for 5 min at 4 °C. The supernatant was subsequently centrifuged again at $11,000 \times g$ for 5 min at 4 °C. ROS was determined by a fluorescent probe DCFH-DA using the ROS assay kit (S0033S; Beyotime, Shanghai, China) based on the manufacturer's directions. After removing the supernatant, the pelleted materials were suspended in incubation buffer (20 mM MOPS, 0.1 M KCl, 10 mM ATP, 10 mM $MgCl_2$, 10 mM sodium succinate, and 1 mM EGTA); then DCFH-DA (1: 1000) was added. Next, the contents were mixed and incubated at 37 °C in the dark for 20 min. Finally, the fluorescence intensity was measured on a flow cytometry with excitation and emission wavelengths of 485 nm and 525 nm, respectively. The results were presented as the fluorescent intensity per nanogram of protein.

## Quantification of MDA and SOD detection
The MDA and SOD activity level of myocardial samples were determined with an MDA test kit (A003-1; Nanjing Jiancheng, Nanjing, China) and SOD test kit (A001-3; Nanjing Jiancheng, Nanjing, China).

## Mitochondrial swelling assay
Myocardial tissue was initially isolated. The tissue was digested using 0.1% collagenase and 0.125% trypsin, followed by hydrolysis at 37°C for 30 min. The mixture was filtered through a 200-mesh screen at 1500 rpm and then centrifuged for 15 min to collect the sediment. mPTP Assay Kit (C2009S; Beyotime, Shanghai, China) was used to determine the opening of the mPTP in cardiomyocytes. This kit provides a more straightforward approach for assessing mPTP opening. In brief, cardiomyocytes were treated with Calcein AM for 30 min at 37°C, which easily entered the cells and accumulated in various

cytosolic compartments, including the mitochondria. Once inside, intracellular esterases broke down the acetoxymethyl esters, releasing the highly polar fluorescent dye calcein. The fluorescence from cytosolic calcein was diminished when $CoCl_2$ was added, while the fluorescence from mitochondrial calcein remained unaffected. However, when the mPTP opened, mitochondrial calcein was released into the cytosol, where it was quenched by $CoCl_2$, leading to a notable decrease in calcein fluorescence. A decrease in absorbance reflects mitochondrial swelling and the opening of the mitochondrial mPTP. Subsequently, flow cytometry was employed to detect the opening of the mPTP in $1 \times 10^6$ cardiomyocytes.

## Detection of myocardial injury

### TUNEL assay

Paraffin-embedded tissue sections were processed with an in-situ cell-death detection kit (11684795910; Roche, Basel, Switzerland) for terminal deoxynucleotidyl transferase-mediated deoxyuridine triphosphate nick-end labeling. The ratio of TUNEL-positive cells to total cells was determined.

### Flow cytometry

To prepare a single-cell suspension, the separation of cardiomyocytes is described as previously mentioned. Cells were washed with PBS, suspended in the binding buffer (from Annexin V-FITC/PI apoptosis detection kit below-mentioned) and stained with AnnexinV-FITC and PI (Annexin V-FITC/PI apoptosis detection kit; A211-02; Vazyme, Nanjing, China). Apoptosis was assessed using flow cytometry.

### Markers of ischemic injury

Blood samples were collected from the femoral artery and centrifuged for 15 min at 3000 rpm to obtain serum. The level of myocardial-injury markers was measured using the CK-MB kit (MM-0625R1; Meimian, Yancheng, China) and the cTn-I kit (MM-0426R2; Meimian, Yancheng, China).

## Histology

Hearts was fixed in 4% paraformaldehyde for 24 hr, dehydrated, rendered transparent, embedded in paraffin and cut into thin sections. After dewaxing and rehydration, the sections were stained with hematoxylin and eosin, and images acquired using a light microscope. A histological score was utilized to evaluate the extent of myocardial injury, based on the criteria established by *Rona et al., 1959*, *Supplementary file 2*.

## Statistics and data analysis

IBM SPSS Statistics software (V26.0, IBM Corp, New York, USA) was used for data analysis. All data are expressed as mean ± standard deviation (Mean ± SD). Sample size for the 4 hr group was calculated as published (*Dell et al., 2002*). The ejection fraction of the first three surviving rats among the groups was measured. The population standard deviation of the variable is 11.34. With a magnitude of difference set at 15, an error rate of 0.1, and a power of 80%, the smallest acceptable sample size was determined to be five animals. Considering an animal loss of approximately 20% after 4 hr of CPR, seven animals were selected for each group. The sample size for the 72 hr group was calculated using StatBox (StatBox-Open 0.1.0; Cloud Powered Clinical Trial, Hebei, China). Set the survival rate at 20% for survival rate 1 and 80% for survival rate 2 (with power = 0.8 and $\alpha$=0.05). Accordingly, 10 was the smallest acceptable animal sample size. The comparison of time-based measurements was evaluated using repeated-measures tow-way ANOVA, with groups as the between-subjects factor and time as the within-subjects factor. One-way ANOVA and Tukey's multiple-comparison test were employed to compare multiple groups. Survival was characterized using the Kaplan-Meier survival analysis test. Statistical significance was defined at $p<0.05$.

## Acknowledgements

We thank LetPub (https://www.letpub.com) for linguistic assistance and presubmission expert review. This work was supported by the National Natural Science Foundation of China (82372210, 81901932) and Natural Science Foundation of Hubei Province (2021CFB492).

# Additional information

### Funding

| Funder | Grant reference number | Author |
| --- | --- | --- |
| National Natural Science Foundation of China | 82372210 | Xiang Zhou |
| National Natural Science Foundation of China | 81901932 | Xiang Zhou |
| Natural Science Foundation of Hubei Province | 2021CFB492 | Xiang Zhou |

The funders had no role in study design, data collection and interpretation, or the decision to submit the work for publication.

### Author contributions
Zhen Wang, Jie Zhu, Writing – original draft; Mengda Xu, Jingyu Yan, Supervision; Xuyuan Ma, Maozheng Shen, Data curation; Guosheng Gan, Xiang Zhou, Writing – review and editing

### Author ORCIDs
Zhen Wang ⓘ http://orcid.org/0009-0007-8947-8885
Maozheng Shen ⓘ http://orcid.org/0009-0008-3750-679X
Jingyu Yan ⓘ https://orcid.org/0009-0000-6905-1146
Xiang Zhou ⓘ https://orcid.org/0000-0001-9635-8741

### Ethics
The experimental protocol was approved by the Animal Experiment Committee of the General Hospital of Central Theater Command (No.2023017) and conformed to the Guide for the Care and Use of Experimental Animals published by the National Institutes of Health, USA (NIH Publication No. 5377-3, 1996). All surgery was performed under pentobarbital sodium anesthesia, and every effort was made to minimize suffering.

Reviewer #1 (Public review): https://doi.org/10.7554/eLife.98554.3.sa1
Reviewer #3 (Public review): https://doi.org/10.7554/eLife.98554.3.sa2
Author response https://doi.org/10.7554/eLife.98554.3.sa3

# Additional files

### Supplementary files
Supplementary file 1. Details of antibodies used in the methodology.

Supplementary file 2. Scoring standard of myocardial pathological injury.

MDAR checklist

### Data availability
All data, except for imaging data that support the findings of this study, have been deposited in Dryad at https://doi.org/10.5061/dryad.9cnp5hqwm. The imaging data have been deposited in BioImages at https://www.ebi.ac.uk/biostudies/bioimages/studies/S-BIAD1760. Source data files have been provided for figures 3 and the figure 6.

The following datasets were generated:

| Author(s) | Year | Dataset title | Dataset URL | Database and Identifier |
|---|---|---|---|---|
| Zhen W, Jie Z, Mengda X, Xuyuan M, Maozheng S, Jingyu Y, Guosheng G, Xiang Z | 2025 | Transplantation of exogenous mitochondria mitigates myocardial dysfunction after cardiac arrest | https://doi.org/10.5061/dryad.9cnp5hqwm | Dryad Digital Repository, 10.5061/dryad.9cnp5hqwm |
| Zhen W, Jie Z, Mengda X, Xuyuan M, Maozheng S, Jingyu Y, Guosheng G, Xiang Z | 2025 | Transplantation of exogenous mitochondria mitigates myocardial dysfunction after cardiac arrest | https://doi.org/10.6019/S-BIAD1760 | Bioimage archive, 10.6019/S-BIAD1760 |

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
