## [Editor Report · eLife Assessment]

In this **valuable** report, the authors investigated the effect of mitochondrial transplantation on post-cardiac arrest myocardial dysfunction (PAMD), which is associated with mitochondrial dysfunction. They **convincingly** demonstrated that mitochondrial transplantation enhanced cardiac function and increased survival rates after the return of spontaneous circulation (ROSC). They have also shown that myocardial tissues with transplanted mitochondria exhibited increased mitochondrial complex activity, higher ATP levels, reduced cardiomyocyte apoptosis, and lower myocardial oxidative stress post-ROSC.

---

## [Referee Report · Reviewer #1 (Public review)]

Summary:

In this study, the authors investigate the effect of mitochondrial transplantation on post-cardiac arrest myocardial dysfunction (PAMD), which is associated with mitochondrial dysfunction. The authors demonstrate that mitochondrial transplantation enhances cardiac function and increases survival rates after the return of spontaneous circulation (ROSC). Mechanistically, they found that myocardial tissues with transplanted mitochondria exhibit increased mitochondrial complex activity, higher ATP levels, reduced cardiomyocyte apoptosis, and lower myocardial oxidative stress post-ROSC.

Strengths:

Previous studies have reported that mitochondrial transplantation can improve myocardial recovery after regional ischemia, but its potential for treating myocardial injury following cardiac arrest has not been tested yet. Therefore, the findings are somewhat novel. Remarkably, the increased survival in mitochondria treated group post ROSC is very promising and highlights its translational potential.

Comments on revisions:

My concerns are adequately addressed.

---

## [Referee Report · Reviewer #3 (Public review)]

In this manuscript titled "Transplantation of exogenous mitochondria mitigates myocardial dysfunction after cardiac arrest", Zhen Wang et al. report that exogenous mitochondrial transplantation can enhance myocardial function and survival rates. It limits mitochondrial morphology impairment, boosts complexes II and IV activity, and increases ATP levels. Additionally, mitochondrial therapy reduces oxidative stress, lessens myocardial injury, and improves PAMD after cardiopulmonary resuscitation. The results of this manuscript clearly demonstrate that mitochondrial transplantation can effectively improve PAMD after cardiopulmonary resuscitation, highlighting its significant scientific and clinical value. The findings shown in this manuscript are interesting to the readers. However, further experiments are needed to confirm this conclusion. In addition, the results should be rewritten to describe and discuss the relevant data in detail.

Major comments from the original round of review:

(1) Can isolated mitochondria be transported to cultured cardiomyocytes, such as H9C2 cells, in vitro?

(2) The description of results in the manuscript is too simple. It lacks detail on the rationale behind the experiments and the significance of the data.

(3) The authors demonstrate that mitochondrial transplantation reduces cardiomyocyte apoptosis. Therefore, Western blot analysis of apoptosis-related caspases could be provided for further confirmation.

(4) Do donor mitochondria fuse with recipient mitochondria? Relevant experiments and data should be provided to address this question.

(5) In Figure 5A, the histograms are not labeled with the specific experimental groups.

Comments on revisions:

The revised manuscript quality has been improved, and most of my concerns were addressed and resolved.

---

## [Author Response]

The following is the authors’ response to the original reviews.

**Reviewer 3 (Public review):**
Major comments:(1) Can isolated mitochondria be transported to cultured cardiomyocytes, such as H9C2 cells, in vitro?

Thank you for this insightful question. Mitochondria are highly dynamic organelles that play a crucial role in cellular energy metabolism. When cells encounter various stressors and increased energy demands, they can benefit from the incorporation of exogenous mitochondria. In 2013, Masuzawa et al. (Masuzawa, et al.,2013) were the first to demonstrate that transplanted mitochondria are internalized by cardiomyocytes 2 to 8 hours after transplantation, significantly contributing to the preservation of myocardial energetics. Ali et al. (Ali, et al.,2020) discovered that exogenous mitochondria could be internalized by H9C2 cardiomyocytes as quickly as 5 minutes after co-incubation, resulting in an acute enhancement of normal cellular bioenergetics following mitochondrial transplantation. Pacak et al. (Pacak, et al.,2015) established that the internalization of mitochondria into cardiomyocytes is time-dependent and occurs through actin-dependent endocytosis.

Collectively, these evidences illustrate that exogenous mitochondria can be effectively internalized by H9C2 cells and other cardiomyocytes, our experiments further confirmed that mitochondrial transplantation can be incorporated by the myocardium in *vivo*.

(2) The description of results in the manuscript is too simple. It lacks detail on the rationale behind the experiments and the significance of the data.

Thank you for this suggestion. We have realized that the results in the submitted manuscript have not been adequately interpreted. We have added necessary details on the rationale behind the experiments and the significance of the data to the results section (Lines 57~59, 69~73, 81~88, 91~98, 100~102, 103~104, 10^9^~115, 124~129, 135~146, 149~157, 159~161, 168~169, 178~179). We would like to express our gratitude to the reviewers once again and hope that our modifications will meet their requirements.

(3) The authors demonstrate that mitochondrial transplantation reduces cardiomyocyte apoptosis. Therefore, Western blot analysis of apoptosis-related caspases could be provided for further confirmation.

Thank you for this constructive comment. We fully agree with the reviewer's perspective on the detection of apoptosis-related caspases and have conducted a Western blot assay to investigate the impact of mitochondria on myocardial tissue. Our new evidence indicates that rats receiving mitochondrial transplantation exhibited reduced expression of cleaved caspase-3 compared with those in the NS and Vehicle groups (Fig. 6G, 6H, Lines 168~169), suggesting that mitochondrial transplantation decreased the level of apoptosis in the myocardium.

(4) Do donor mitochondria fuse with recipient mitochondria? Relevant experiments and data should be provided to address this question.

This is a very helpful comment. Investigating the fate of transplanted mitochondria in myocardial cells after CA is of great significance. The internalization of exogenous mitochondria has been observed across various cell types (Liu, et al.,2021; Shanmughapriya, et al.,2020). Notably, a recent study indicated that after being incorporated into host cells, isolated mitochondria are transported to endosomes and lysosomes. Subsequently, most of these mitochondria escape from these compartments and fuse with the endogenous mitochondrial network (Cowan, et al.,2017). We have discussed this in the manuscript. (Lines 217~220)

Oxidative stress, a pathophysiological phenomenon common to cells suffering from ischemia/reperfusion insults after CA/CPR, was implicated to promote internalization and survival of exogenous mitochondria (Aharoni-Simon, et al.,2022). In our study, we confirmed that mitochondrial transplantation can enhance the metabolism of cardiomyocytes, increase ATP level, and reduce reactive oxygen species (ROS). Our results indirectly confirm that isolated mitochondria can successfully fuse with myocardial mitochondria.

(5) In Figure 5A, the histograms are not labeled with the specific experimental groups.

We apologize for this oversight. We have labeled the specific experimental groups in the histograms presented in Figure 6B and 6C (originally Figure 5A).

**Reviewer #1 (Recommendations For The Authors):**
(1) The age, gender, and strain of the donor rats should be specified in the Methods section. Additionally, it is not obvious what doses of mitochondria were injected into the rats and how the dosage was initially determined.

Thanks for your suggestion. We have included relevant information about the donor rats in the Methods section（Lines 361~362）.

In Mito group, each animal received 0.5 mL of 1× 10^9^/mL mitochondrial suspension. (Lines 342~345). Considerable amounts of data have demonstrated the efficacy of mitochondrial transplantation in cellular, animal, and human research (Alemany, et al.,2024; Kaza, et al.,2017; Liu, et al.,2023). However, there is currently no evidence to determine the optimal dosage for transplantation. In previous research, isolated mitochondria (1 × 10^9^) were delivered to the left coronary ostium in pigs, and can be a viable treatment modality in cardiac ischemia-reperfusion injury (Blitzer, et al.,2020; Guariento, et al.,2020). Additionally, the dose of 1× 10^9^ mitochondria achieve the maximal hyperemic effect when administered via intracoronary injection (Shin, et al.,2019). Considering that Sprague-Dawley (SD) rats are smaller than pigs and that there is a loss of mitochondria during pulmonary circulation, we adopted a mitochondrial transplantation dose of 5× 10^8^. We will explore the optimal dosage in our future research.

(2) In Figure 4a, the number of transplanted mitochondria appears to be very low. Considering the high number of mitochondria present in cardiomyocytes, it is unclear whether this small amount of transplanted mitochondria can significantly impact complex II activity and ATP levels in myocardial tissues, as shown in Figures 4b-d, or improve survival post-ROSC, as shown in Figure 2d. Could the observed benefits of mitochondrial transplantation be due to the indirect effects of the injected mitochondria, such as the release of mitochondrial contents, rather than the mitochondria themselves, as discussed by Bertero et al. (2021, Circ. Research)? This issue should be addressed in the manuscript.

Thanks for this wonderful comment. As presented in Fig. 4 (originally Figure 4A), our results indicated the internalization of mitochondria by myocardium, shown by colocalization of Mito-tracker and myocardium marker. We would like to make our points here regrading to Fig. 4:

(1) Significant left ventricular systolic and diastolic dysfunction that occurs in the myocardium shortly after the return of ROSC is referred to post-cardiac arrest myocardial dysfunction (PAMD) (Laurent, et al.,2002). It has demonstrated the efficacy of mitochondrial transplantation for the heart following ischemia-reperfusion injury in cellular, animal, and human studies, despite inadequate mitochondrial internalization (Liu, et al.,2023). A low number of transplanted mitochondria may improve cardiac function.

(2) Only biologically active mitochondria can be specifically labeled with Mito-tracker. Therefore, cardiomyocytes uptake mitochondria that possess complete functionality. Previous results have demonstrated that mitochondrial contents, such as nonviable mitochondria, mitochondrial fractions, mitochondrial deoxyribonucleic acid, ribonucleic acid, exogenous adenosine diphosphate and ATP, do not provide protection to the ischemic heart (McCully, et al.,2017; McCully, et al.,2009).

(3) The specific mechanism for mitochondrial internalization has yet to be fully elucidated. We totally agree with reviewer’s opinion pertaining the presence of other mechanisms of mitochondria transplantation that play a role in cardiac protection. Multiple mechanism may involve in the cardiac protection effect of mitochondria transplantation, and we are actively seeking reasonable approach to verify these hypotheses in an underway study (Lines 236~246).

(3) In Figure 4g, the claims regarding sarcomere length, mitochondrial structure, the number of cristae, accumulated calcium etc. seem to rely on the visual interpretation of representative images. To ensure a reliable interpretation of the data, a blinded quantification of each image in each group should be conducted. The same applies to the claims made in Figure 5E.

Thanks for this suggestion. We have quantitatively evaluated the electron microscope images and HE images of the myocardium to ensure reliable interpretation. Corresponding supplements have been added to the methods (Lines 433~441, 494~496), results sections (Lines 10^9^~115, 178~179), and Figures 5C, 5D, 6K and 6H (originally Figures 4G and 5E).

(4) In line 69, it is unclear why the authors claim that MAP and HR decrease at 1, 2, 3, and 4 hours after ROSC in all groups compared to the Sham group, despite stating in line 72 that "MAP and HR did not differ at any observational time points (P>0.05, Figure 2C)."

We apologize for our inaccurate phrasing. In the presented study, there was no statistically significant difference between MAP and HR at any observational timepoints (P>0.05, Figure 2C). In the NS, Vehicle and Mito groups, the MAP and HR decreased at 1, 2, 3, and 4 hours after ROSC, reaching their nadir at 1 hour. Subsequently, MAP and HR increased gradually but did not show any statistically significant differences compared with the Sham group. (Lines 69~73).

(5) The absence of increased mitochondrial content in the mito-groups should be discussed further in the manuscript.

Thank you for your suggestion. We discussed the reasons why the mass of isolated mitochondria did not increase in Lines 224~235.

(6) The N in Figure 5d should be provided.

Thanks for your suggestion. We have revised the figure legend to include N of Figure 6F (originally Figures 5D).

(7) Figure 6 demonstrates content beyond the findings in this manuscript. This reviewer recommends limiting the graphical abstract to the findings specifically in this paper.

Thanks for your great advice. We have revised Figure 7 (originally Figure 6) and restricted the graphical abstract to the findings presented in this paper.

Minor issues:(8) The order of data in Figure 4 should be consistent with the text in the manuscript. Figures 4E-F-G are described before Figures 4B-C-D in the text. Similarly, Figure 5F was described before Figure 5E in the text.

Thanks for your great advice. We have rearranged the order of the pictures to align with the text. Thank you for your proposal.

(9) In Figure 4A, the locations of the epicardium, muscle, and endocardium should be indicated for clarity. Also, it is not obvious where the close-up box refers to in the actual image.

Thank you for your suggestion. We primarily seek evidence of mitochondrial internalization within the endocardium, as injury occurs first during myocardial ischemia (Kuwada and Takenaka,2000). The close-up box in Fig. 4 refers to the endocardium.

(10) In Figure 5A, the group annotations are missing from the MDA and SOD graphs. The standard deviation bars for the SOD vehicle and SOD mito groups (3rd and 4th columns) appear to overlap. Can the authors provide the actual p-values?

We apologize for the mission of group annotations in the MDA and SOD graphs. The p-value between the Vehicle group and the Mito group was 0.004. The SOD activity level of myocardial samples in the groups are presented in Table 1.

**Author response table 1. sa3table1:** The SOD activity levels of myocardial samples in groups (U/mgprot).

	Sham	NS	Vehicle	Mito
n	7	7	7	7
Mean +- SD	5.55+-0.35	4.02+-0.15	4.34+-0.45	5.00+-0.22

(11) In line 58, NS abbreviation is used without defining what NS is.

We apologize for not including the full name of NS. NS is the abbreviation of normal. It has now been marked in the manuscript. (Line 58)

(12) In line 118, what MDA stands for is not described until line 348. MDA should be defined in the text for the general audience.

We apologize for this. We have defined it in the manuscript. (Lines 156~157)

(13) In line 192, the authors state that "mitochondrial transplantation... increased the expression of antioxidant enzymes after four hours of ROSC," while only SOD activity levels were assessed in the manuscript. Increased activity levels do not necessarily imply an increase in expression levels. This discrepancy should be addressed in the Discussion section.

Sorry for confusing the ‘activity’ with ‘expression’. Although mitochondrial transplantation has been shown to be involved in the restoration of manganese superoxide dismutase levels after ischemic insults, the changes in antioxidant enzyme expression level were not evaluated at the protein level in this paper (Tashiro, et al.,2022). To avoid misunderstandings, we have replaced the term ‘expression’ with ‘activity’ as appropriate. (Lines 268~271)

(14) Mitochondria from non-ischemic gastrocnemius muscle of health donor animals were isolated and a manner that maximized their healing potential. This sentence is not clear.

We apologize for the confusing sentence in the original manuscript. To improve clarity, we have revised that sentence. We isolated mitochondria from allogeneic gastrocnemius muscle tissue of healthy rats and maintained optimal mitochondrial activity and therapeutic effects. (Lines 199~201)

Minor grammar issues:In line 153, mitochondrial should be mitochondria.Figure 2D: Percent servival should be percent survival.There should be a blank in complex IIactivity Figure 4B, and complex IV activity in Figure 4C.In line 134, Four hours of ROSC, Tissue samples from. Tissue is capital.In line 190, Similaerly should be similarly.

Thank you for your valuable comments. We apologize for the grammatical issues caused by our oversight. We have made the necessary corrections in the manuscript and figures. (Lines 198, 179, and 268), Figure 2D, Figure 5E (originally Figure 4B); Figure 5F (originally Figure 4C).

**Reviewer #2 (Recommendations For The Authors):**
Some details are lacking clarity, such as the rationale behind choosing certain doses or time points for interventions.

Thank you for this valuable suggestion. We have explained the rationale behind the selection of the dosage and the timing of the intervention. (Lines 201~212)

I would suggest verifying mitochondrial function using the seahorse experiment oxygen consumption, and to check mitochondrial oxidative stress. I would also suggest checking the mitochondrial permeability transition pore opening, using for example calcein cobalt quenching or simply a kit to examine this further.

Thank you for your valuable advice. In our manuscript, we added results regarding mitochondrial reactive oxygen species (ROS) and the mitochondrial permeability transition pore (mPTP) opening. As anticipated, mitochondrial transplantation reduced the increase in mitochondrial ROS and the mPTP opening in ischemic myocardium. (Lines 135~146, 149~157, 442~455, 460~476, Figure 5H, 5I, 6A)

We agree that seahorse experiment oxygen consumption would be beneficial for understanding the intricacies of their interactions and enhancements. Additionally, Ali et al. (Ali, et al.,2020) have demonstrated that introducing non-autologous mitochondria from healthy skeletal muscle cells into normal cardiomyocytes results in a short-term improvement in bioenergetics, as measured using a Seahorse Extracellular Flux Analyzer. In our results, we have not yet conducted cellular experiments, The process of isolating cells from the myocardial tissue of adult SD rats for Seahorse analysis can lead to secondary damage to the myocardial cells (Jacobson, et al.,1985). In this experiment, we measured ATP content and the activity of mitochondrial complexes to evaluate energy changes after mitochondrial transplantation. We will conduct cell experiments and utilize Seahorse measurements to further clarify the alterations in myocardial energy in future.

For Figure 3B, it would be beneficial to include the relative quantification of the mitochondrial marker COX-IV. Additionally, if feasible, I suggest verifying the representation of the mitochondria outer membrane TOM20 or VDAC.

Thank you for your great suggestion. As suggested, we added TOM20 to assess the purity of the isolated mitochondria and reached the same conclusion: the isolated mitochondria exhibited high purity (Figure 3B). TOM20 was expressed in both muscle lysates and isolated mitochondria, whereas GAPDH was exclusively found in the muscle lysate. (We re-validated the purity of the mitochondria by using relative quantification of TOM20 and COX VI.)

In Figure 2C, the clarity of the graphs depicting both arterial pressure (MAP) and heart rate (HR) is lacking and could potentially confuse the reader. I recommend incorporating color coding instead of relying solely on symbols, or by presenting the data in a more comprehensible format and that aligns with graph B as well.

Thank you for your constructive comments. We have color-coded the diagrams in Figure 2B and 2C.

In Figure 4A, please include high-magnification of the mitochondria to provide a more detailed examination.

Thank you for this insightful comment. We have provided a high-magnification image of the mitochondria in Figure 4.

Regarding lines 81-82, I recommend specifying the sentence more precisely for better clarity and understanding.

Thank you for your comments. We have revised the sentences in lines 83~86 to enhance their clarity for readers.

In the Materials and Methods section, it is crucial to provide precise details. For instance, when staining the exogenous mitochondria with MitoTracker Red, it is important to specify the duration of staining, such as the standard 20 minutes for example. Additionally, it is advisable to mention the number of times these mitochondria were washed with the respiratory solution to ensure thorough removal of excess MitoTracker, thus preventing unintended staining of endogenous mitochondria with MitoTracker red upon injection of pre-labeled mitochondria.

Thank you for your suggestion. We have added the necessary details regarding Mito-Tracker Red dyeing. (Lines 373~376) In addition, we also added other details in necessary (Lines 373~376, 379~382, 395~396, 397~400, 487~488). We appreciate your suggestion once again.

The sensitivity of JC-1 dye to temperature and pH fluctuations underscores the necessity for meticulous experimental conditions. It is crucial for the authors to elucidate why they chose to maintain the samples at 4 {degree sign} C for 60 minutes, especially considering the dye's optimal operating temperature of 25 {degree sign} C. Providing a rationale behind this deviation from standard protocol would enhance the scientific rigor and reproducibility of the study. Please add more information on the objectives used in the fluorescence microscope (BX53, OLYMPUS, Tokyo, Japan) and the software used.

We sincerely apologize for the mistake in this sentence. The purified mitochondria, which are stained with JC-1, should be stored at 4°C and examined using a fluorescence microscope within 60 minutes. Purified mitochondria were incubated with JC-1 staining solution at 37°C for 20 minutes. The fluorescence microscope used in our experiment is equipped with a WHN 10/22 eyepiece, and the software version is OLYMPUS cellSens Standard 3.2. (Lines 379~382)

Moreover, in the context of immunoblotting, it is imperative for the authors to furnish detailed information regarding the preparation of muscle tissue homogenates. Specifically, clarification is needed regarding the solution utilized for tissue grinding. Did the authors employ ice-cold RIPA lysis buffer or an alternative lysis buffer, supplemented with a protease inhibitor cocktail? Such details are pivotal for methodological transparency.

Thanks for this wonderful comment. In the methods section, we added detailed information about protein extraction. (Lines 383~385)

Furthermore, it would be beneficial for the authors to specify the instrument employed for scanning the immunoblots, as well as the software utilized for subsequent analysis of the immunoblot images. Providing this information would not only enhance the reproducibility of the findings but also facilitate the evaluation of the experimental results.

Thank you for your suggestion. We have included the instrument used for scanning the Western blot, as well as the software used for image analysis in the manuscript. (Lines 397~400)

Authors must exercise caution against copy-pasting. In line 282, there's a query regarding how the mitochondria were isolated. It is recommended to cite a specific reference and offer more comprehensive details. Despite the authors referencing a number within the text, the absence of numbered references makes it challenging to cross-reference.

Thank you for pointing this out; we have updated the citation accordingly (Line 361).

Figure 5C please double check some misspelling label errors (e.g: Vehicle and not Vehucle).

We apologize for the misspelling in Figure 6E (originally Figure 5C) and have corrected it. Additionally, we have thoroughly reviewed the text for spelling errors and sincerely apologize once again for the previous mistakes. (Lines 249~252, 322)

References:

Aharoni-Simon M, Ben-Yaakov K, Sharvit-Bader M, Raz D, Haim Y, Ghannam W, Porat N, Leiba H, Marcovich A, Eisenberg-Lerner A, Rotfogel Z. 2022. Oxidative stress facilitates exogenous mitochondria internalization and survival in retinal ganglion precursor-like cells. *SCI REP-UK* 12:5122. doi:10.1038/s41598-022-08747-3

Alemany VS, Nomoto R, Saeed MY, Celik A, Regan WL, Matte GS, Recco DP, Emani SM, Del NP, McCully JD. 2024. Mitochondrial transplantation preserves myocardial function and viability in pediatric and neonatal pig hearts donated after circulatory death. *J THORAC CARDIOV SUR* 167: e6-e21. doi: 10.1016/j.jtcvs.2023.05.010

Ali PP, Kenney MC, Kheradvar A. 2020. Bioenergetics Consequences of Mitochondrial Transplantation in Cardiomyocytes. *J AM HEART ASSOC* 9: e14501. doi:10.1161/JAHA.119.014501

Blitzer D, Guariento A, Doulamis IP, Shin B, Moskowitzova K, Barbieri GR, Orfany A, Del NP, McCully JD. 2020. Delayed Transplantation of Autologous Mitochondria for Cardioprotection in a Porcine Model. *ANN THORAC SURG* 109:711-719. doi: 10.1016/j.athoracsur.2019.06.075

Cowan DB, Yao R, Thedsanamoorthy JK, Zurakowski D, Del NP, McCully JD. 2017. Transit and integration of extracellular mitochondria in human heart cells. *SCI REP-UK* 7:17450. doi:10.1038/s41598-017-17813-0

Guariento A, Blitzer D, Doulamis I, Shin B, Moskowitzova K, Orfany A, Ramirez-Barbieri G, Staffa SJ, Zurakowski D, Del NP, McCully JD. 2020. Preischemic autologous mitochondrial transplantation by intracoronary injection for myocardial protection. *J THORAC CARDIOV SUR* 160: e15-e29. doi: 10.1016/j.jtcvs.2019.06.111

Jacobson SL, Banfalvi M, Schwarzfeld TA. 1985. Long-term primary cultures of adult human and rat cardiomyocytes. *BASIC RES CARDIOL* 80 Suppl 1:79-82. doi:10.1007/978-3-662-11041-6_15

Kaza AK, Wamala I, Friehs I, Kuebler JD, Rathod RH, Berra I, Ericsson M, Yao R, Thedsanamoorthy JK, Zurakowski D, Levitsky S, Del NP, Cowan DB, McCully JD. 2017. Myocardial rescue with autologous mitochondrial transplantation in a porcine model of ischemia/reperfusion. *J THORAC CARDIOV SUR* 153:934-943. doi: 10.1016/j.jtcvs.2016.10.077

Kuwada Y, Takenaka K. 2000. [Transmural heterogeneity of the left ventricular wall: subendocardial layer and subepicardial layer]. *J CARDIOL* 35:205-218.

Laurent I, Monchi M, Chiche JD, Joly LM, Spaulding C, Bourgeois B, Cariou A, Rozenberg A, Carli P, Weber S, Dhainaut JF. 2002. Reversible myocardial dysfunction in survivors of out-of-hospital cardiac arrest. *J AM COLL CARDIOL* 40:2110-2116. doi:10.1016/s0735-1097(02)02594-9

Liu D, Gao Y, Liu J, Huang Y, Yin J, Feng Y, Shi L, Meloni BP, Zhang C, Zheng M, Gao J. 2021. Intercellular mitochondrial transfer as a means of tissue revitalization. *SIGNAL TRANSDUCT TAR* 6:65. doi:10.1038/s41392-020-00440-z

Liu Q, Liu M, Yang T, Wang X, Cheng P, Zhou H. 2023. What can we do to optimize mitochondrial transplantation therapy for myocardial ischemia-reperfusion injury? *MITOCHONDRION* 72:72-83. doi: 10.1016/j.mito.2023.08.001

Masuzawa A, Black KM, Pacak CA, Ericsson M, Barnett RJ, Drumm C, Seth P, Bloch DB, Levitsky S, Cowan DB, McCully JD. 2013. Transplantation of autologously derived mitochondria protects the heart from ischemia-reperfusion injury. *AM J PHYSIOL-HEART C* 304:H966-H982. doi:10.1152/ajpheart.00883.2012

McCully JD, Cowan DB, Emani SM, Del NP. 2017. Mitochondrial transplantation: From animal models to clinical use in humans. *MITOCHONDRION* 34:127-134. doi: 10.1016/j.mito.2017.03.004

McCully JD, Cowan DB, Pacak CA, Toumpoulis IK, Dayalan H, Levitsky S. 2009. Injection of isolated mitochondria during early reperfusion for cardioprotection. *AM J PHYSIOL-HEART C* 296:H94-H105. doi:10.1152/ajpheart.00567.2008

Pacak CA, Preble JM, Kondo H, Seibel P, Levitsky S, Del NP, Cowan DB, McCully JD. 2015. Actin-dependent mitochondrial internalization in cardiomyocytes: evidence for rescue of mitochondrial function. *BIOL OPEN* 4:622-626. doi:10.1242/bio.201511478

Shanmughapriya S, Langford D, Natarajaseenivasan K. 2020. Inter and Intracellular mitochondrial trafficking in health and disease. *AGEING RES REV* 62:101128. doi: 10.1016/j.arr.2020.101128

Shin B, Saeed MY, Esch JJ, Guariento A, Blitzer D, Moskowitzova K, Ramirez-Barbieri G, Orfany A, Thedsanamoorthy JK, Cowan DB, Inkster JA, Snay ER, Staffa SJ, Packard AB, Zurakowski D, Del NP, McCully JD. 2019. A Novel Biological Strategy for Myocardial Protection by Intracoronary Delivery of Mitochondria: Safety and Efficacy. *JACC-BASIC TRANSL SC* 4:871-888. doi: 10.1016/j.jacbts.2019.08.007

Tashiro R, Bautista-Garrido J, Ozaki D, Sun G, Obertas L, Mobley AS, Kim GS, Aronowski J, Jung JE. 2022. Transplantation of Astrocytic Mitochondria Modulates Neuronal Antioxidant Defense and Neuroplasticity and Promotes Functional Recovery after Intracerebral Hemorrhage. *J NEUROSCI* 42:7001-7014. doi:10.1523/JNEUROSCI.2222-21.2022